# FLAM: Scaling Latent Action World Models with Factorization

## Abstract

Learning latent actions from action-free video has emerged as a powerful paradigm for scaling up controllable world models learning. The latent actions offer an extra degree of freedom for users to generate videos iteratively. However, existing approaches often rely on monolithic inverse and forward dynamics models to learn one latent action that controls all, which struggle to scale in complex scenes where different entities act simultaneously. In this work, we propose FLAM, a factored dynamics framework that decomposes the latent state into independent factors, each with its own inverse and forward dynamics model. This structure enables more accurate modeling of complex, multi-entity dynamics and improves the video generation quality in action-free video settings. Evaluated on Multigrid, Procgen, nuPlan, Sports and EGTEA datasets, FLAM consistently outperforms the monolithic dynamics model, demonstrating the superiority of the factorized model.

## 1 Introduction

Recent advances in Latent Action Models (LAM) (Schmidt & Jiang, 2023; Bruce et al., 2024) have unlocked the possibilities of learning world models from action-free videos that are abundant on the web. Specifically, these approaches use an inverse dynamics model to encode environmental changes into a *single* latent action. The latent action is then used to train a forward dynamics model, allowing controllable predictions of future frames purely from in-the-wild videos.

However, in-the-wild videos often contain complex scenes where many entities may be taking actions simultaneously: for instance, a robot video may include independent joint movements, object manipulation, and shifting camera perspectives; while a soccer game involves several players, the ball, and even background audience motion, each of which acts independently. Compressing all these motions into a *single* latent action is challenging, since the complexity of actions scales exponentially with the number of movable entities (Fig. 1). Consequently, existing methods struggle with latent action learning in such settings, which severely limits their scalability to in-the-wild scenarios.

In this work, we propose the Factored Latent Action Model (FLAM), where the latent state is decomposed into a set of factors, each independently predicting its latent action and its next state via shared factored inverse and forward dynamics models. By sharing a common latent action codebook across factors, FLAM reduces the challenge of learning a massive codebook covering all action combinations to the simpler task of learning a small codebook for each entity's action. Unlike most prior LAM approaches that use a monolithic scene representation entangling all entities, FLAM factorizes the scene into compositional entities with a shared forward dynamics model, inherently supporting permutation invariance and enabling stronger generalization. With next frame prediction as the training objective, FLAM learns structured state and action representations from action-free video data, leading to more accurate modeling of complex, multi-entity dynamics and improved prediction quality compared to previous work.

We evaluated our method in challenging domains including real-world autonomous driving, sports, and egocentric activity videos, where FLAM outperforms previous state-of-the-art methods both in dynamics prediction quality and video generation controllability.

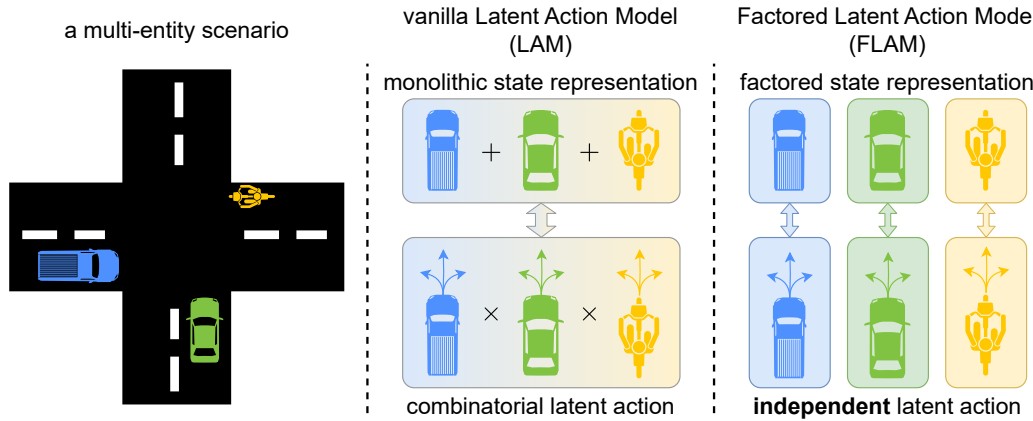

Figure 1: In multi-entity scenarios, **(left)** such as an intersection with three road users: **(middle)** a vanilla latent action model must compress all possible action combinations into a single latent action, which makes learning challenging due to the need for a huge codebook of size $|\mathcal{A}|^K$. **(right)** In contrast, FLAM decomposes the state and latent representation into $K$ factors and adopts the same small action codebook of size $|\mathcal{A}|$ for each factor, enabling more efficient learning.

## 2  RELATED WORK

In this section, we first discuss previous works in learning world models from action-free videos. Next, we discuss prior works in factorized decision making and object-centric representations, whose strength and weakness inspire FLAM.

**Dynamics Modeling without Action Labels**   Given the abundant source of videos and the scarcity of action labels, several methods have been developed to learn from pure observations. PlaNet (Hafner et al., 2019) learns a latent dynamics model for planning. Minderer et al. (2019) learns both object structure and dynamics from videos using keypoint representation. ILPO (Edwards et al., 2019) learns a latent policy with forward dynamics model only, and later maps the latent policy output to real action through an action remapping network. LAPO (Schmidt & Jiang, 2023) and Genie (Bruce et al., 2024) jointly learn an inverse dynamics model jointly with a forward dynamics model. Ye et al. (2024) expand learning from observation from vision only to vision and language modalities. Past work Zhang et al. (2022); Ye et al. (2022); Baker et al. (2022); Ye et al. (2024) has proven the value of dynamics modeling without action labels in applications such as learning to drive, play games, and manipulate with robot arms from videos. Our work is mainly based on the LAPO framework, since it is the SOTA method that incorporates all four components: inverse dynamics modeling, forward dynamics modeling, real action remapping and policy learning. However, noisy real-world videos usually include more than one entity, and LAPO falls short in modeling those complex scenes, which motivates our method focusing on the dynamics modeling for multi-entity videos through factorization.

**Factorization in Decision Making**   Factorization has long been used to exploit structured state and action spaces in complex environments, often through the factored MDP formulation (Osband & Van Roy, 2014; Guestrin et al., 2003). Recent works have applied this principle to derive factored forward dynamics (Pitis et al., 2020; Wang et al., 2022), factored policies (Hu et al., 2025; 2024), and factored value functions (Sodhani et al., 2022). FLAM is motivated by the same principle that factorization simplifies complexity into manageable components. However, it applies this idea to learning world models from action-free video, where the underlying factorization is not given.

**Object-Centric Representation Learning**   Object-centric learning aims to represent complex scenes by isolating individual objects from the background and from each other, leading to improved generalization and modeling capabilities. Key challenges include the need for supervision, leading to the development of many unsupervised methods. MONet(Burgess et al., 2019) and IO-DINE(Greff et al., 2019) achieved unsupervised multi-object segmentation through attention and

iterative amortized inference. Slot Attention (Locatello et al., 2020) uses iteratively updated slots to learn disentangled, object-based representations. DINO (Caron et al., 2021; Oquab et al., 2023) uses a transformer-based architecture instead. Recent work such as Chen et al. (2021); Jiang et al. (2023) has focused on improving object-centric learning fidelity in noisy real-world data. In our work, we use Slot Attention as the factorizer for the state representation and use the forward prediction task as the learning signal. One thing that distinguishes our representation from the object-centric representation is that we do not use the reconstruction task for representation learning; thus the representation tends to focus on gathering groups of entities with similar dynamics instead of those with similar superficial visual properties.

## 3 PRELIMINARIES

**Latent Action Model**  Given a video dataset, we seek to model the dynamics with only observations; without any action labels. The Latent Action Model (LAM) is a model that uses an inverse dynamics model (IDM) and a forward dynamics model (FDM) to infer latent actions. Though not fully aligned with real actions with physical meanings, the inferred latent actions capture the most essential change during the transition, thus can be used for subsequent policy learning.

As shown in Fig. 2, both the IDM and the FDM observe $o_t$, but only the IDM observes $o_{t+1}$. Therefore, in order to accurately predict $o_{t+1}$, the IDM must extract useful transition information through the latent action $a_t$ for the FDM. However, if no bottleneck constraints are added in $a_t$, the model is likely to collapse, as a naive solution would be simply to copy $o_{t+1}$.

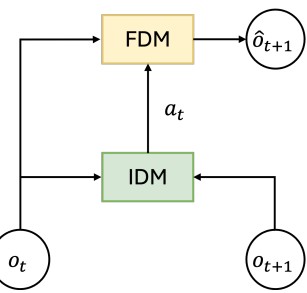

Thus, prior methods use vector quantization to discretize the latent action into a small set of codes, ensuring that it is compact (Van Den Oord et al., 2017). While action quantization is a necessary step to ensure that it only encodes the most important changes rather than copying the future frame, it also limits the expressiveness of the latent action. One example is scenarios where many state variables have independent actions, such as a crowded intersection, and it is suboptimal and even infeasible to compress all entities' actions into a single latent action.

Figure 2: Latent action model architecture.

As a solution, FLAM factorizes both the state representation and the latent action into independent factors, enabling us to share a small codebook across all factors, as discussed in detail in Section 4.

## 4 FACTORED LATENT ACTION MODEL (FLAM)

From a high-level perspective, FLAM scales latent action models to multi-entity scenarios by inferring a set of latent actions rather than a single latent action between each pair of frames in a factored manner, enabled by two learning phases shown in Fig. 3:

- **Encoder learning** (Section 4.1): FLAM pre-trains a VQ-VAE to extract high-level features from pixels, allowing the latent action model to learn in the feature space rather than the pixel space for the purpose of efficient learning.

- **Latent action model (LAM) learning** (Section 4.2): Using the extracted features, FLAM decomposes the scene into several independent factors, also referred to as *slots*. For each slot, an inverse dynamics model infers a separate latent action from its current and next-frame values. Then, based on its current value and the corresponding latent action, a forward dynamics model independently predicts the next-frame value for each slot. Finally, all predicted slots are mapped back to the feature space and decoded into the next video frame.

After FLAM is well trained to model the world from action-free videos, its latent action outputs can be used for either controllable video generation or policy learning to achieve tasks. We refer to this as the third phase, and discuss how to leverage FLAM for both settings in Section 4.3.

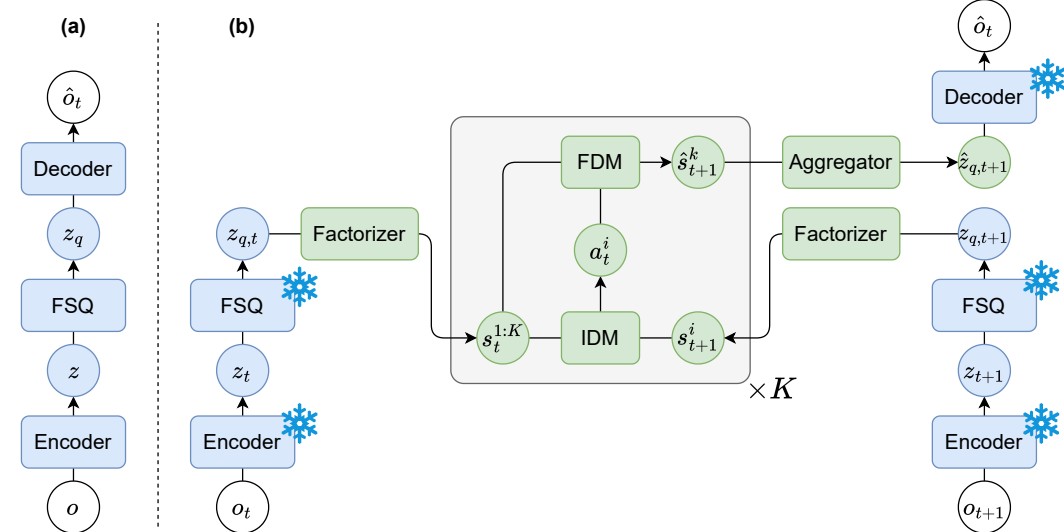

Figure 3: Two training stages of FLAM. **(a)** A VQ-VAE is pretrained to extract features for latent action model learning. **(b)** FLAM infers latent actions and makes predictions for each slot independently, with all modules trained to minimize prediction error.

## 4.1 PRETRAINED ENCODER

As shown in Fig. 3 (a), FLAM learns a VQ-VAE (Van Den Oord et al., 2017) to extract features from raw pixels, enabling fast LAM learning. For each frame $o \in \mathbb{R}^{H \times W \times 3}$, a CNN encoder first extracts $N$ patch-level features $z \in \mathbb{R}^{N \times d_z}$. The features are then quantized from continuous encoder outputs to the nearest entry in a discrete codebook, denoted as $z_q \in \mathbb{R}^{N \times d_z}$, using finite scalar quantization FSQ (Mentzer et al., 2023), which forces features into a finite set of learned embeddings. At the same time, we also record their codebook indices as $c \in \{1, \ldots, C\}^N$. By applying vector quantization, FLAM reformulates the next-step feature prediction from a regression task (predicting features $z$) to a classification task (predicting indices $c$), which has been shown to improve prediction quality (Singh et al., 2021) (see details in Sec. 4.2). Finally, a decoder reconstructs the frame from the quantized feature:

$$z = \texttt{Encoder}(o),$$
$$z_q, c = \texttt{FSQ}(z),$$
$$\hat{o} = \texttt{Decoder}(z_q).$$

The VQ-VAE is trained by minimizing the following reconstruction loss:

$$\mathcal{L}_{\texttt{VQ-VAE}}(o) = ||o - \hat{o}||^2. \tag{1}$$

## 4.2 FACTORED LATENT ACTION MODEL (FLAM)

As shown in Fig. 3 (b), our latent action model contains four key components: 1) a factorizer that decomposes the scene $z_q$ into a set of independent slots $s$, 2) an inverse dynamics model that infers a separate latent action $a_t^i$ for each slot, 3) a forward dynamics model that, given the current slot value and latent action, predicts the next-frame value for each slot, and 4) an aggregator that maps the predicted slots back to the feature space. These four components are jointly trained to minimize the feature prediction error.

**Factorizer** To decompose the scene into a set of factors with independent actions, FLAM uses slot attention (Locatello et al., 2020). For each frame, $K$ slots $s_t \in \mathbb{R}^{K \times d_s}$ are initialized from learned embeddings and then compete to bind to different regions in the frame through iterative slot attention:

$$s_t = \texttt{SlotAttention}(z_{q,t}). \tag{2}$$

---

**Algorithm 1** FLAM Latent Action World Model

---

1: Prepare a dataset of $(o_t, o_{t+1})$.
2: Initialize the VQ-VAE (`Encoder`, `FSQ`, `Decoder`) and the latent action model (`Factorizer`, `IDM`, `FDM`, `Aggregator`).
3: Pretrain the VQ-VAE with the reconstruction loss in Eq. (1).                    ▷ Sec. 4.1
4: // Train the latent action model                    ▷ Sec. 4.2
5: Extract features with the VQ-VAE: $z_{q,t} = \text{FSQ}(\text{Encoder}(o_t))$, $z_{q,t+1} = \text{FSQ}(\text{Encoder}(o_{t+1}))$.
6: Extract slots using Eq. 2: $s_t^{1:K} = \text{Factorizer}(z_{q,t})$, $s_{t+1}^{1:K} = \text{Factorizer}(z_{q,t+1})$.
7: **for** each slot $i = 1, \ldots, K$ **do**
8:     Infer the latent action: $a_t^i = \text{IDM}(s_t^{1:K}, s_{t+1}^i)$.
9:     Predict the next-frame slot: $\hat{s}_{t+1}^i = \text{FDM}(s_t^{1:K}, a_t^i)$.
10: Map predicted slots back to the feature space: $\hat{z}_{q,t+1} = \text{Aggregator}(\hat{s}_{t+1}^{1:K})$.
11: Optimize the latent action model with the prediction loss in Eq. 5.

---

Note that although FLAM employs slot attention, a technique that is also widely used for learning object-centric representations, FLAM differs from prior object-centric representation work in its learning signal. While object-centric representation methods typically optimize reconstruction and extract objects based on visual features, FLAM instead optimizes prediction and extracts factors based on action independence, as detailed in the Training Objective paragraph.

**Inverse Dynamics Model (IDM)**    After extracting independent slots, the IDM aims to infer a latent action $a_t^i$ for each slot $i$, based on all current slots $s_t^{1:K}$ and its next-frame value $s_{t+1}^i$:

$$a_t^i = \text{IDM}(s_t^{1:K}, s_{t+1}^i), \tag{3}$$

where for a variable $x$, $x^{1:K}$ denotes the set $\{x^i\}_{i=1}^K$, and we use $s_t^{1:K}$ and $s_t$ interchangeably. We use all current slots rather than just $s_t^i$ as inputs to account for interactions between factors, enabling more accurate latent action prediction (e.g., a person in a car moves because of the car rather than by themselves).

To implement the IDM, we adopt the spatio-temporal model introduced in Genie (Bruce et al., 2024). The spatial block applies self-attention to capture interaction information across $s_t^{1:K}$, while the temporal block applies cross-attention to compare $s_{t+1}^i$ with its current value and encode the most meaningful changes between them:

$$s_t^{1:K} = \text{SelfAttention}(s_t^{1:K}),$$
$$a_t^i = \text{CrossAttention}(\text{query} = s_{t+1}^i, \text{key} = [s_t^i, s_{t+1}^i]).$$

Note that although we only use the current and next values in the temporal block, our model can be easily extended to use all prior frames $s_{1:t}^i$ as inputs.

Similar to existing LAM methods, we use vector quantization to compress the latent action into a small set of codes. This compression limits the capacity of the latent action, preventing it from simply copying $s_{t+1}^i$ and bypassing dynamics learning. In our work, we use the same IDM (including the latent action codebook) for all slots.

**Forward Dynamics Model (FDM)**    To provide the learning signals for the IDM, the forward dynamics model takes all current slots $s_t^{1:K}$ together with the latent action $a_t^i$ and predicts the next-frame value for each slot $\hat{s}_{t+1}^i$:

$$\hat{s}_{t+1}^i = \text{FDM}(s_t^{1:K}, a_t^i). \tag{4}$$

Similarly to the IDM, we use all current slots instead of just $s_t^i$ as inputs to capture interactions between slots, as they are not intended to be encoded by the latent action. For implementation, we use the same spatio-temporal model as the IDM, except that the temporal block uses $a_t^i$ as the key. We also use the same FDM to predict all slots.

**Aggregator** Finally, the aggregator maps the predicted slots back to the feature space so they can be decoded into the next-frame prediction. As mentioned in Sec. 4.1, the features extracted by the VQ-VAE are selected from a codebook, and their values can be predicted with their codebook indices. Prior work has empirically shown that predicting indices leads to better image quality than predicting feature values directly (Singh et al., 2021). Specifically, the aggregator predicts the feature indices in an autoregressive manner. For each image patch $n$, it first generates a categorical distribution over the codebook indices $\sigma_{t+1}^n$, conditioned on all prior patches and slots using cross-attention. It then selects the feature from the codebook using the index with the highest probability:

$$\sigma_{t+1}^n = \texttt{CrossAttention}(\texttt{query} = \hat{z}_{q,t+1}^{n-1}, \texttt{key} = [\hat{z}_{q,t+1}^{1:n-1}, \hat{s}_{t+1}^{1:K}]),$$
$$\hat{z}_{q,t+1}^n = \texttt{ExtractCode}(\arg\max \sigma_{t+1}^n),$$

where, when predicting the first patch, we use a learnable embedding for $\hat{z}_{t+1}^0$.

**Training Objective** The four components of the latent action model (Factorizer, IDM, FDM, and Aggregator) are trained jointly to minimize the feature prediction error, expressed as the cross-entropy between the predicted categorical distribution of codebook indices and the ground truth, summed across all $N$ image patches:

$$\mathcal{L}_{\texttt{LAM}}(z_{q,t}, z_{q,t+1}, c_{t+1}) = \sum_{n=1}^{N} \texttt{CrossEntropy}(\sigma_{t+1}^n, c_{t+1}^n). \tag{5}$$

As mentioned earlier, although the factorizer and the aggregator use implementations similar to prior object-centric representation methods, the prediction loss, together with the design of independent latent actions for each slot in the IDM and FDM, enables FLAM to separate entities in a scene according to their independent dynamics. This design differs from object-centric representation methods, which rely on reconstruction loss and separate objects based on visual appearance.

### 4.3 LEARNED LATENT ACTIONS UTILIZATION

The latent actions learned from FLAM implicitly incorporate the action information of each entity in the world, thus offering a degree of freedom to manipulate the video generation. We can let human specify a latent action value from the learned latent action codebook, and use that as a control variable to generate video accordingly. Conditioned on different latent action chosen, the video generation will be diverse.

The latent actions can also be used as action labels for policy learning on video frames. Because the dimensions of the latent action do not necessarily align with the dimensions of the real action, a latent action decoder $f$ that maps latent action $a$ to real action $u$ is needed. We first collect a small dataset of latent action-action pairs and train a latent action decoder $f$ offline through supervised learning. We can then learn a policy $\pi_u$ through behavior cloning on the observation-action pairs $(o, f(a))$ obtained by applying FLAM and a latent action decoder on expert demonstration videos $D_E$:

$$\mathcal{L}_{\pi_u} = \mathbb{E}_{(o,a) \sim D_E} \left\| \pi_u(o) - f(a) \right\|^2. \tag{6}$$

## 5 EXPERIMENTS

Our central hypothesis is that our factorized representation and latent action learning approach can better capture the features and dynamics of each entity separately, thus leading to more accurate world modeling on complex, noisy, multi-entity videos. These more accurate world models can then be used for planning and policy learning, leading to higher performance on downstream tasks. To this end, we evaluate our method in both clean simulation settings as well as more noisy real-world settings. We focus on designing our experiments to answer the following key questions:

1. Does enforcing discretization in the prediction output truly help avoid error accumulation?

2. Does factorization truly help learning more accurate world models?

3. Can FLAM work well on noisy real-world data?

## 5.1 SETTINGS

We conducted experiments on 2 simulation datasets and 3 real-world datasets. The simulation datasets include a Multigrid dataset and a Procgen dataset. The real-world datasets include an autonomous driving dataset, a sports dataset, and an egocentric activity dataset. We pick these datasets because they all include multiple independent entities, such as cars, players, or two human arms. Please refer to Appendix B for more details. The hyperparameters are listed in Tab. 4.

## 5.2 BASELINES

We compare our method with three baseline algorithms. Latent Action Model (LAM) is the vanilla variant that learns latent actions without factorization. LAM using Object-Centric representation (OC+LAM) is the variant that adds a reconstruction loss on top of the prediction loss of the vanilla LAM. World Model (WM) is the variant without factorization and uses ground truth actions rather than latent actions for forward dynamics prediction. While LAM and LAM+OC run on all datasets, WM only runs on the simulation datasets because ground-truth action labels are not available within the real-world datasets.

## 5.3 ENCODER-DECODER PRE-TRAINING

As introduced in Section 4, the encoder-decoder architecture feature extractor is pre-trained and frozen in the latent action model learning stage. Instead of training a universal feature extractor for all data, we separately train a feature extractor for each dataset. This is because the focus of our work is to prove the superiority of factorization in world modeling, rather than a universal world model. Feature extractors customized for data can ensure that this part is not the bottleneck of our model, for the sake of fair comparison of multiple latent action model variants. Fig. 5 and Fig. 6 in Appendix C are the illustrations of the original video frames and the reconstructed video frames using the trained feature extractor. We can tell from the small difference between the original frames and reconstructed frames that the feature extractors are trained very well to capture useful information.

## 5.4 HOW DOES FACTOR AGGREGATION TYPE AFFECT DYNAMICS PREDICTION ACCURACY?

As mentioned in Section 4.2, we used a patch-by-patch auto-regressive transformer aggregator in FLAM, and multi-step prediction in both LAM and FLAM, therefore the error in (patched) representation is likely to accumulate and degrade the performance of the entire world model. Instead of predicting the continuous value of the representation in the next time step (regression mode), we predict the codebook indices of the quantized representation at next time step (classification mode), taking advantage of discretization to avoid error accumulation.

Though prior work Singh et al. (2021) has shown empirically that predicting indices leads to better image quality than predicting feature values directly, here we train and test both the regression version and classification version of the Multigrid dataset, an example result is shown in Appendix D as Fig.7. As we can see, the model that predicts the codebook indices outperforms the model that predicts the representation value. Our hypothesis is verified; the classification type of aggregation indeed helps improve the world modeling.

## 5.5 DO WE NEED FACTOR TRACKING?

When using slot attention as the factorizor and inferring a separate latent action for each slot, a question would naturally arise that whether we need one-on-one matching between slots on two different frames. We tried multiple ways of slot matching and find that recursive slot initialization - using the final slots of previous frame as the initialization for iterative slot updates on current frame would lead to the best results. This reveals that it is too hard for the factorized world model to implicitly learn the tracking of factors and there is the necessity of offering priors through explicit slot matching. For details of various slot matching methods we tried and results comparison, please refer to Appendix E.

## 5.6 HOW DOES FLAM PERFORM COMPARED TO THE VANILLA LATENT ACTION MODEL?

Based on the findings in Section 5.4, we predict the feature indices in all the following experiments. We examine the performance of FLAM from two perspectives, the dynamics modeling accuracy, i.e. how good the model is in future video frame prediction, and controllability, i.e. how much adjustment users can apply onto video generation through the latent actions. For dynamics modeling accuracy, after scaling the pixel values into the [0,1] space, we calculate the Peak Signal-to-Noise Ratio (PSNR) as the evaluation metric, using the mean square error (MSE) between the 1-step ahead ground-truth frame and predicted frame:

$$\text{PSNR}_{t=1}(x_t, \hat{x}_t) = 10 * \log_{10}\left(\frac{1}{\text{MSE}(x_t, \hat{x}_t)}\right),$$

where $x_t$ represents the ground-truth frame at time $t$ and $\hat{x}_t$ represents the predicted frame at time $t$ using latent actions $a_{1:t}$ inferred from ground-truth frames. The bigger the PSNR is, the more accurate the model is on dynamics modeling. For controllability, we use a metric called the PSNR difference devised by Bruce et al. (2024):

$$\Delta_t \text{PSNR} = \text{PSNR}_t(x_t, \hat{x}_t) - \text{PSNR}_t(x_t, \hat{x}_t').$$

It measures the difference between the video frames $\hat{x}_t$ generated conditioned on a sequence of latent actions inferred from ground-truth frames, and the video frames $\hat{x}_t'$ generated conditioned on a sequence of latent actions randomly sampled from a categorical distribution. The greater $\Delta_t \text{PSNR}$, the higher level of controllability the latent actions offer during the video generation.

The results are shown in Tab. 1. The results show that FLAM achieves lower predection error than other baselines without access to ground truth actions. This validates our assumption that in environments with multiple independent entities, factorizing the states and learning a dynamics model separately for each entity would help modeling the world dynamics more accurately. Although in Procgen environment, FLAM gets slightly worse controllability than LAM. This leads us to carefully choose the number of factors in the following experiments, considering the complexity of the scenes. Meanwhile, because FLAM outperforms OC+LAM, we can claim that without learning object-centric representation through an auxiliary reconstruction task, we can still do world modeling well by learning dynamics-aware representations, especially in the multi-entity settings. In fact, OC+LAM appears to suffer from a performance degradation compared to LAM, which reveals that object-centric representations are not always beneficial to world modeling, if learned from superficial visual appearance. Due to space limit, please refer to Appendix F for visualized examples of reconstruction and prediction rollout samples.

Table 1: Video prediction performance of all methods across the simulation datasets.

|  | Dataset | WM | LAM | OC+LAM | FLAM |
|---|---|---|---|---|---|
| PSNR(↑) | Multigrid | 48.90 | 31.04 | 26.79 | **35.49** |
|  | Procgen | 24.74 | 25.28 | 20.87 | **26.28** |
| $\Delta_t$ PSNR(↑) | Multigrid | - | 7.84 | 0.11 | **11.24** |
|  | Procgen | - | **5.02** | 2.18 | 3.99 |

## 5.7 WHEN DO WE NEED FLAM?

To investigate whether FLAM indeed help with dynamics modeling in complex multi-entity videos, we conduct an ablation study over the number of entities in the multigrid environment. As we can see from 2, when the number of moving entities increases in the scene, FLAM is more robust than non-factored LAM.

Table 2: Video prediction performance across Multigrid environments with various number of entities.

|  | # Entities | WM | LAM | FLAM |
|---|---|---|---|---|
|  | 1 | 57.46 | 32.04 | **35.39** |
|  | 2 | 56.56 | 33.39 | **34.47** |
| PSNR($\uparrow$) | 4 | 48.90 | 31.04 | **35.49** |
|  | 8 | 39.85 | 26.18 | **28.36** |
|  | 16 | 34.03 | 24.61 | **25.24** |

## 5.8 IS FLAM ROBUST TO REAL-WORLD SCENES?

We now apply FLAM to real-world datasets, with results shown in Tab.3. On all three dataset, FLAM outperforms OC+LAM which confirms the significance of learning dynamics-aware representations instead of visual representations. While FLAM and LAM shows similar capability in accurate prediction, FLAM demonstrate superior controllability in video generation. Due to space limitations, please refer to Appendix F for visualized examples of reconstruction and prediction rollout samples.

Table 3: Video prediction performance of all methods across the real-world datasets.

|  | Dataset | LAM | OC+LAM | FLAM |
|---|---|---|---|---|
|  | nuPlan | **18.55** | 16.59 | 17.70 |
| PSNR($\uparrow$) | Sports | **17.96** | 15.68 | 17.47 |
|  | EGTEA | **19.41** | 13.47 | 19.35 |
|  | nuPlan | 2.07 | 0.26 | **5.97** |
| $\Delta_t$ PSNR($\uparrow$) | Sports | 1.58 | 0.63 | **4.94** |
|  | EGTEA | 4.58 | 0.73 | **5.13** |

## 5.9 CONTROLLABLE VIDEO GENERATION

The factored latent actions learned by FLAM not only help with accurate world modeling, but can also serve as a manipulation surrogate to guide the video generation. After an action codebook is learned, we let human user specify a choice from the codebook, and then rollout multiple steps. As shown in Fig. 4, latent actions can be used as a control variable to generate various video frames even with the same initial frame.

## 6 CONCLUSION

We present FLAM, a factored latent action model that scales to multi-entity scenarios where the action space grows exponentially with the number of entities. Although the large action space makes learning the action codebook challenging for vanilla LAM, FLAM addresses this issue by decomposing the state and the latent action representation into multiple independent factors, allowing learning to focus on a small action codebook shared across factors. This structure enables us to learn structured state and action representations from action-free video data, leading to more accurate modeling of complex multi-entity dynamics and improved both prediction quality and video controllability.

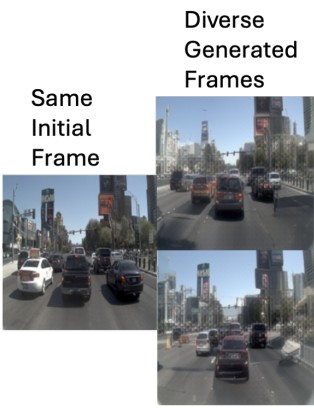

Figure 4: Conditioned on different latent actions, diverse videos can be generated on same initial frame.

FLAM makes predictions in latent space with a transformed-based aggregator and a pre-trained decoder for visualization. For future work, exploring more sophisticated decoding methods, such as diffusion models, could further enhance the visual quality of generated rollouts.

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

## A    HYPERPARAMETER

Table 4: Experiment hyperparameters. Simulation datasets are Multigrid and Procgen; real-world datasets are Sports, EGTEA, and nuPlan.

|  | **Simulation** | **Real-world** |
|---|---|---|
| Encoder | IMPALA | MAGVIT-v2 |
| Decoder | LAPO | MAGVIT-v2 |
| Image size | 128 (Multigrid), 224 (Procgen) | 224 |
| Tokenizer quantizer | FSQ | FSQ |
| Tokenizer codebook size | 1024 | 16384 |
| Tokenizer codebook levels | [4,4,4,4,4] | [4,4,4,4,4,4,4] |
| $z$-channels | 128 | 128 |
| Codebook dimension | 128 | 128 |
| $d_{\mathrm{model}}$ (LAM/FLAM) | 256 / 256 | 256 / 512 |
| Number of blocks | 2 | 3 |
| Number of factors $K$ | 4 (Multigrid), 32 (Procgen) | 16 (nuPlan), 32 (Sports & EGTEA) |
| Action quantizer | FSQ | FSQ |
| Action codebook size | 4 (Multigrid), 16 (Procgen) | 32 |
| Action FSQ levels | [2,2] or [2,2,2,2] | [2,2,2,2,2] |
| Sub-trajectory length | 10 | 10 |
| Prediction steps | 5 | 5 |
| Hierarchical subtokens | [32,32] | [128,128] |

## B    DATASET DETAILS

We summarize here the datasets used in our experiments with additional implementation details. All datasets are split into train/validation/test with an 80-10-10 ratio by frame count, unless otherwise specified. Frames are scaled to $[0, 1]$ and resized to the image resolutions reported in Table 4.

**Multigrid (Simulation).**    We extend the MiniGrid environment (Chevalier-Boisvert et al., 2023) to support multiple independently moving agents. Each video consists of $32 \times 32$ gridworlds rendered at $128 \times 128$ resolution, with between 2–4 agents moving simultaneously. The number of factors $K$ is set to 4 to match the number of independent agents. We use the official `empty-agent_K-v0` (train) and `v1` (validation) splits from the Minari dataset format.

**Procgen (Simulation).**    We adopt the Procgen benchmark (Cobbe et al., 2019) with background rendering disabled for consistency. We use 16 training games and 100 held-out levels, following the configuration used in Schmidt & Jiang (2023). Videos are rendered at $224 \times 224$ resolution. We set the number of factors $K = 32$ to capture the diverse moving entities.

**nuPlan (Real-world).**    We use the nuPlan benchmark (Caesar et al., 2022), restricting to front-facing camera streams only. Videos are downsampled from 10 fps to 5 fps (frame skip = 2). Frames are resized to $224 \times 224$, and the number of factors is set to $K = 16$ to capture vehicles and pedestrians in each scene. We use the `train_CAM_F0.hdf5` and `valid_CAM_F0.hdf5` splits.

**Sports (Real-world).**    The sports dataset combines five sub-datasets: (1) soccer, basketball, and volleyball sequences from SportsMOT (Cui et al., 2023); (2) volleyball data from (Ibrahim et al., 2016); (3) tennis videos from Playable Video Generation (Menapace et al., 2021); (4) the TenniSet dataset (Faulkner & Dick); and (5) basketball clips from BASKET (Pan et al., 2025). All videos are sampled at 5 fps (frame skip = 3 from original 15 fps) and resized to $224 \times 224$. Because multiple players are present in each scene, we set $K = 32$.

**EGTEA (Real-world).**    We additionally evaluate on EGTEA, an egocentric video dataset of hand-object interactions. Following prior work, we use the official train/validation split with frame skip 2 and resize frames to $224 \times 224$. We set $K = 32$ to reflect the diversity of object-level factors.

## C EXAMPLES OF VIDEO FRAME RECONSTRUCTION OF THE ENCODER-DECODER FEATURE EXTRACTOR

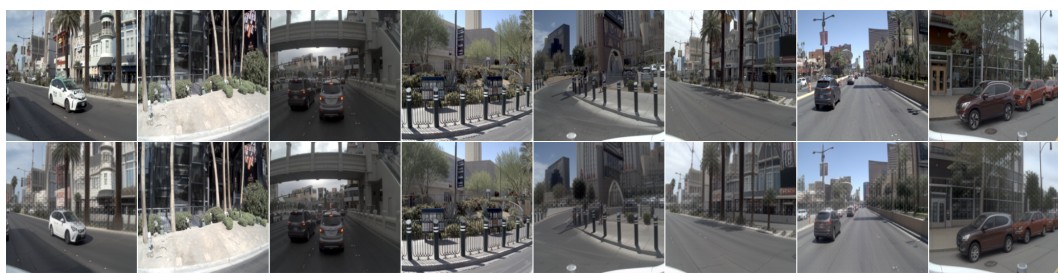

Figure 5: The reconstruction results on nuPlan dataset. The images on top are the original frames, the images at the bottom are reconstructed images using the extracted features.

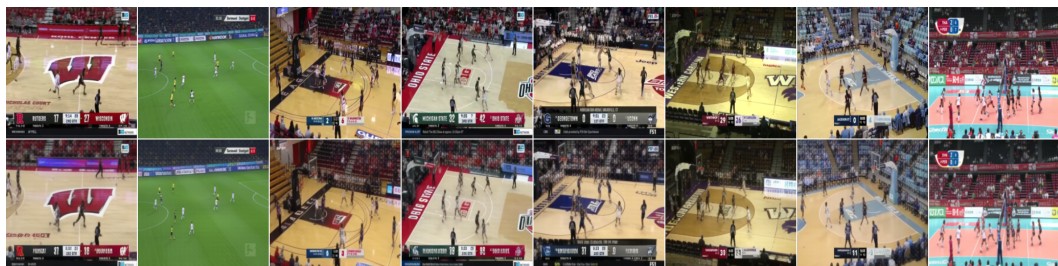

Figure 6: The reconstruction results on Sports and dataset. The images on top are the original frames, the images at the bottom are reconstructed images using the extracted features.

## D    COMPARISON OF RESULTS USING PREDICTION TYPE OF REGRESSION AND CLASSIFICATION

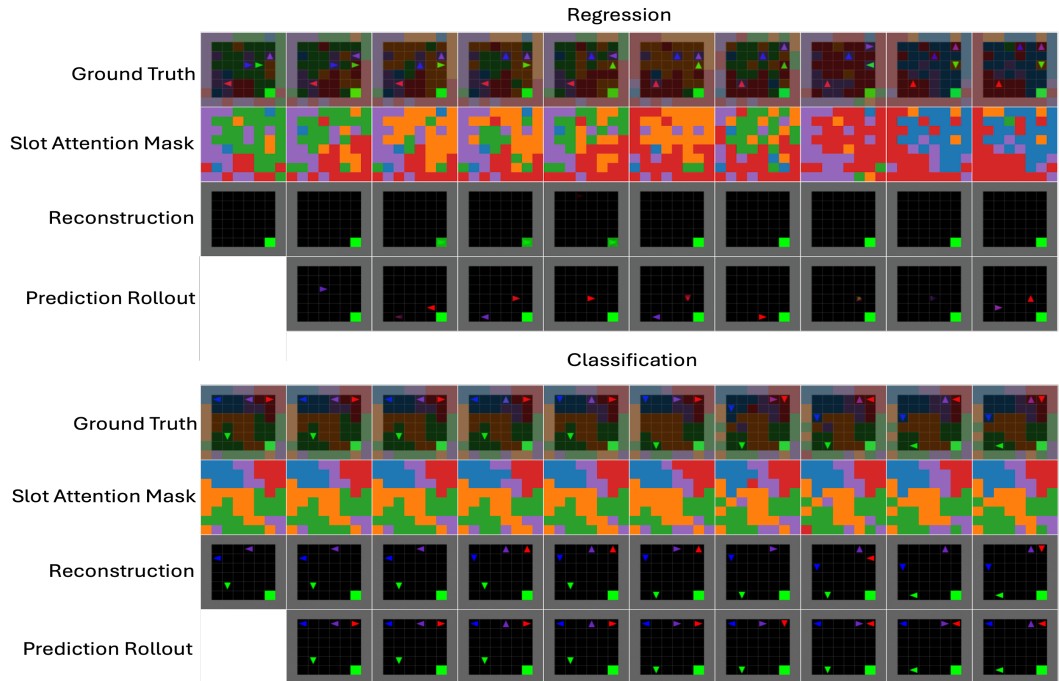

Figure 7: Example of reconstruction and prediction rollout results on Multigrid dataset using prediction type of regression and classification. The first row is the ground truth frames from the dataset. The second row is the attention mask from the factorizor module, in which each color represent a factor(visualization color is randomly assigned to each factor). The third row is the reconstructed frames and the fourth row is the prediction frames.

## E    EXPLICIT SLOT MATCHING

While slot attention is good at factorizing the state features into $K$ slots, each attending to a subarea in the frame, there is no guarantee of entity tracking. To perform separate dynamics learning for each slot-based factor is highly based on the assumption that the slot $s_t^k$ at time $t$ and slot $s_{t+1}^k$ at time $t+1$ we used for each IDM and FDM are corresponded to the same entity (subarea). Therefore, we tried two categories of slot matching, one is slot consistency loss and the other is recursive slot initialization.

Because there is no constraint to enforce the same distribution for the slot representations of different frames, which can cause problem for the following dynamics learning, we applied a slot distribution loss

$$\mathcal{L}_{\texttt{slot\_dist}} = D_{KL}(s_t^{1:K}\|s_{t+1}^{1:K})$$

to encourage the slot representations across consecutive frames follow the same distribution. To enforce the consistent ordering of the slots, we also tried hard 1-to-1 Hungarian matching with MSE loss and greedy top-1 matching with contrastive infoNCE loss. For Hungarian matching, we find the permutation $\sigma^*$ (one-to-one assignment) that minimizes total cost:

$$\sigma^* = \arg\min_{\sigma \in \mathfrak{S}_K} \sum_{i=1}^{K} C_{i,\sigma(i)},$$

where $\mathfrak{S}_K$ is the set of permutations of $\{1, \ldots, K\}$, and $C_{i,\sigma(i)}$ is the distance between the paired slots $s_t^i$ and $s_{t+1}^{\sigma(i)}$. Then Hungarian matching loss is the matched cost:

$$\mathcal{L}_{\texttt{slot\_Hungarian}} = \sum_{i=1}^{K} C_{i,\sigma^*(i)}.$$

For greedy top-1 matching, we treat each $s_t^i$ as a query, and use $s_{t+1}^j$ for all $j$ as keys. Then top-1 slot matching loss for slot $s_t^i$ is:

$$\mathcal{L}_i = -\log\left(\frac{\exp\big(\text{sim}(s_t^i, s_{t+1}^{j^*})/\tau\big)}{\sum_{j=1}^{K}\exp\big(\text{sim}(s_t^i, s_{t+1}^j)/\tau\big)}\right),$$

where $j^*$ is the index of the most similar slot (top-1 match):

$$j^* = \arg\max_j \text{sim}(s_t^i, s_{t+1}^j).$$

Then contrastive matching loss is

$$\mathcal{L}_{\texttt{slot\_contrastive}} = \frac{1}{K}\sum_{i=1}^{K}\mathcal{L}_i.$$

We augment the FLAM training objective $\mathcal{L}_{\text{LAM}}$ with each of the slot consistency loss above, but there is no significant improvement in the accuracy of future frame prediction. Therefore, we use recursive slot initialization instead. For the slot attention learning at time $t$, we initialize the slot embeddings with the values from slots of previous time $t-1$ rather then random initialization, and then start the iterative slot updates. The results in Fig. 8 show that recursive slot initialization helps learn better slots, thus we use this implementation as the default slot initialization method for our FLAM model.

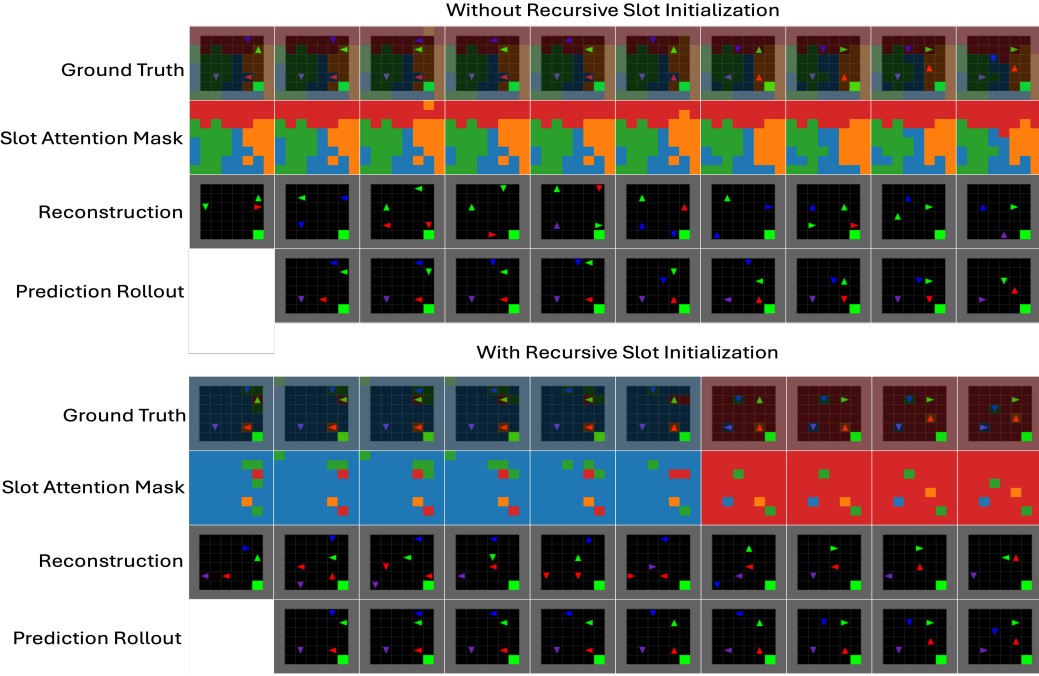

Figure 8: Comparison of results with and without recursive slot initialization on Multigrid dataset. The first row is the ground truth frames from the dataset. The second row is the attention mask from the factorizor module, in which each color represent a factor. The third row is the reconstructed frames,

## F  EXAMPLES OF RECONSTRUCTION AND PREDICTION ROLLOUT RESULTS

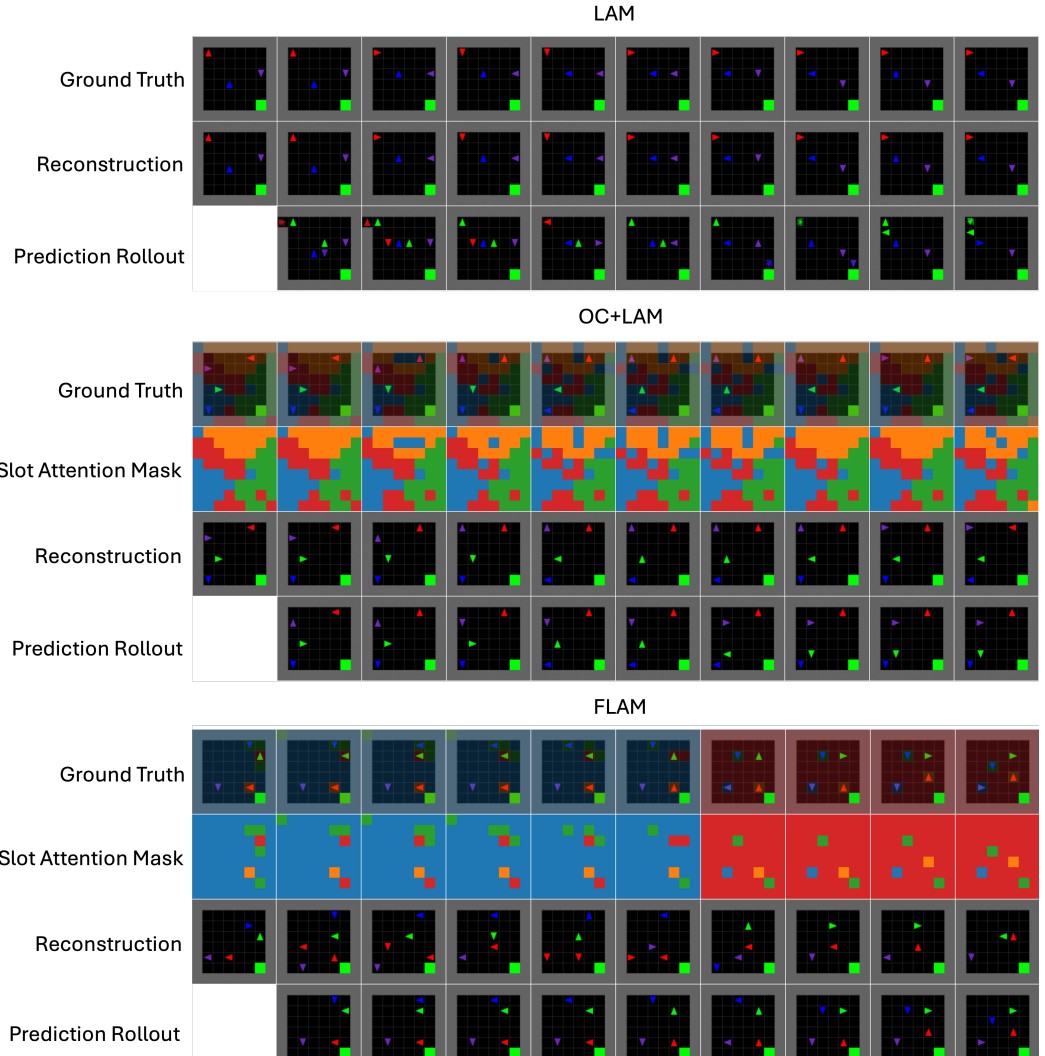

Figure 9: Example of reconstruction and prediction rollout results of LAM, OC+LAM and FLAM on Multigrid dataset.

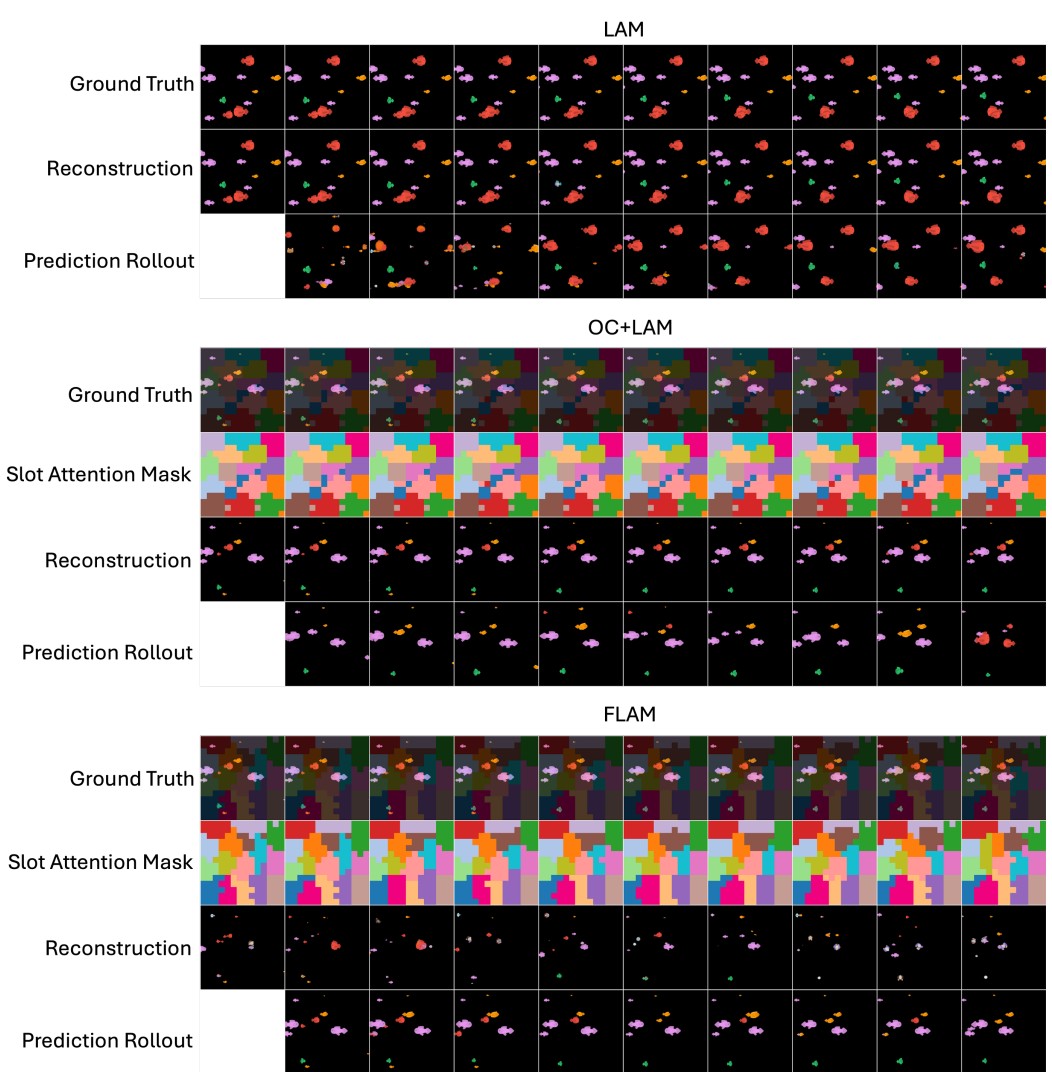

Figure 10: Example of reconstruction and prediction rollout results of LAM, OC+LAM and FLAM on Procgen dataset.

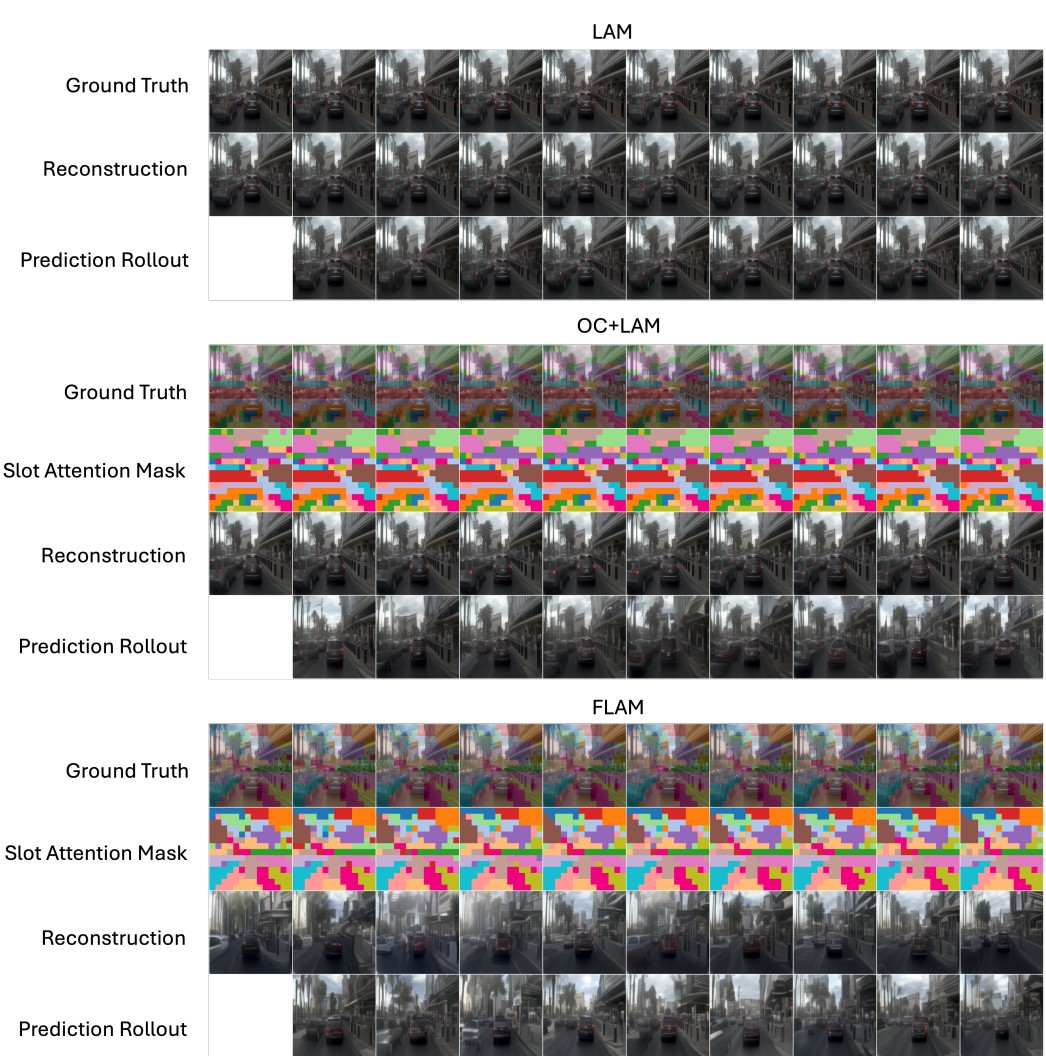

Figure 11: Example of reconstruction and prediction rollout results of LAM, OC+LAM and FLAM on nuPlan dataset.

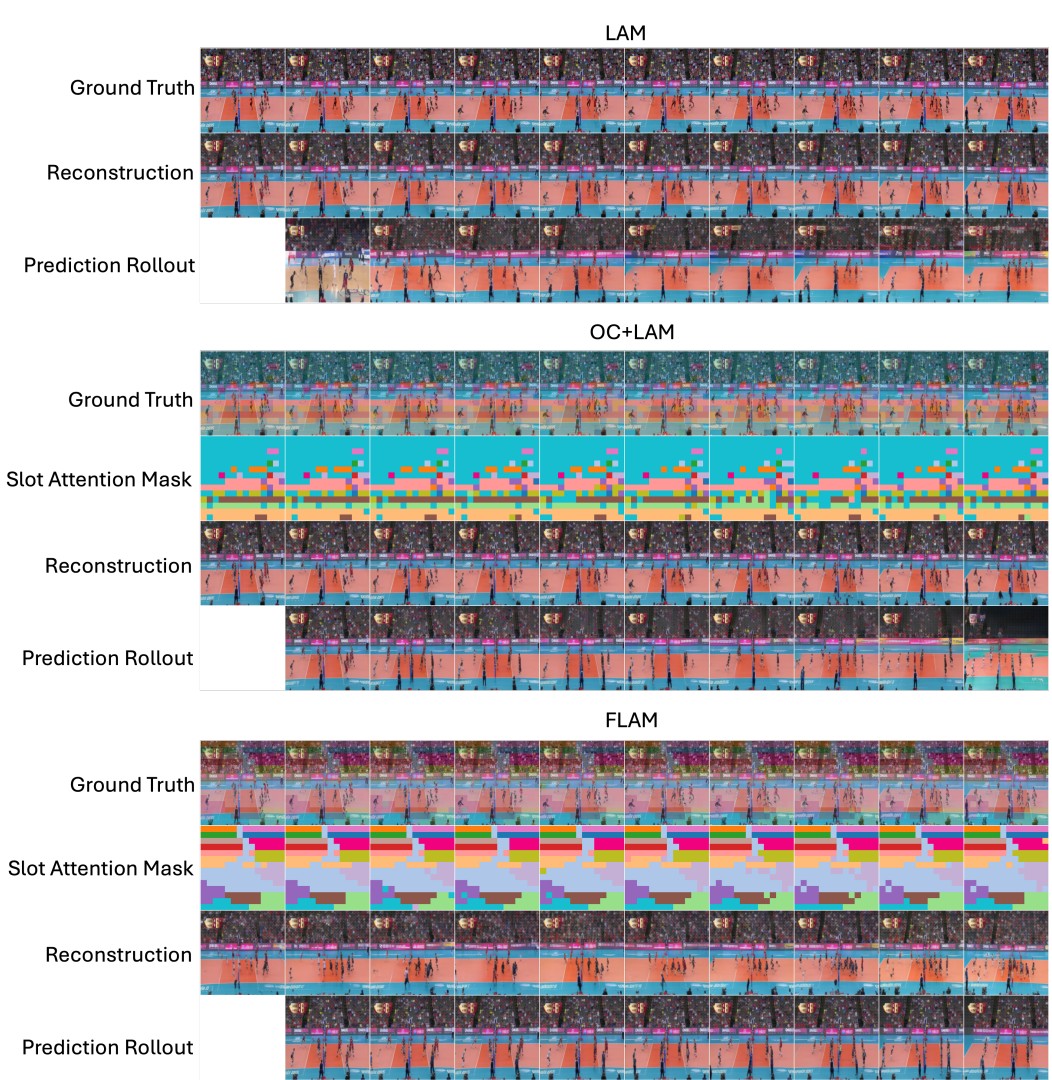

Figure 12: Example of reconstruction and prediction rollout results of LAM, OC+LAM and FLAM on Sports dataset.

