# OpenReview forum: "FLAM: Scaling Latent Action World Models with Factorization"
_ICLR.cc/2026/Conference — Submitted to ICLR 2026_

### Official Review · Reviewer_2C1D · 2025-10-31

**Soundness:** 3
**Presentation:** 1
**Contribution:** 2
**Rating:** 2
**Confidence:** 5

**Summary:**

This paper proposes a Factorized Latent Action Model (FLAM) that decomposes both latent representations and latent actions into multiple factors, aiming to enhance the modeling capability of latent action world models in complex multi-entity scenarios. The approach leverages slot attention to obtain object-centric latent representations and applies shared inverse and forward dynamics networks to each component. Experiments are conducted on Multigrid, Procgen, nuPlan, Sports, and EGTEA datasets.

**Strengths:**

1. The idea of object-centric decomposition for latent action models is novel and, to the best of my knowledge, has not been explored before.
2. The paper includes extensive experiments across five datasets, covering both synthetic and real-world scenarios, which demonstrates strong empirical effort.

**Weaknesses:**

1. **Major concern: not good presentation and missing experimental evidence**
   The main text lacks critical experimental results needed to substantiate the paper’s claims. Several key findings are either deferred to the appendix or missing entirely.
   - The results shown in Figure 7, which support a major conclusion of the paper, should appear in the main text. Moreover, quantitative ablation results should accompany this figure to strengthen the argument.
   - The paper mentions that “human users” were asked to specify latent actions, but it does not clarify which latent actions (e.g., action indices or semantic descriptions) were selected for the visualization in Figure 4.
   - Section 4.3 mentions the use of action labels for policy learning, but no corresponding experiments are presented.
   - The paper repeatedly claims (Lines 113, 233, 291) that the learned object-centric representations group entities with similar dynamics rather than visual appearance, yet no qualitative evidence is provided beyond overall quantitative metrics. I strongly suggest including visual showcases illustrating this property.
   - Although not an experimental issue, the discussion on slot initialization in Section 5.5 is insightful and should be moved into the method section, which contributes more directly to understanding the model design.
   - Overall, I strongly recommend a thorough revision of the paper’s structure and formatting to make the contributions clearer and ensure that all major claims are properly supported.
2. The title emphasizes “scaling latent action world models,” yet the experiments do not demonstrate any scaling trend with model size or capacity (see also Question 3). To avoid misleading implications, the paper should either include such experiments or adopt a more conservative title reflecting its actual scope.

**Questions:**

1. A straightforward baseline could be to factorize latent actions into multiple groups, analogous to how DreamerV2 decomposes latent representations. How would FLAM compare to such a baseline?
2. The number of slots is known to be a critical hyperparameter in slot attention. Have the authors conducted sensitivity analyses on the number of slots or other key hyperparameters?
3. The proposed FLAM yields lower PSNR on real-world datasets. Could the authors explain why? Does the factorization of latent actions inherently limit representational capacity or generalization?

---

> ### Author Response · Authors · 2025-11-16
> **Response to reviewer's comment**
>
> > The results shown in Figure 7, which support a major conclusion of the paper, should appear in the main text. Moreover, quantitative ablation results should accompany this figure to strengthen the argument.
>
> We agree that more intuitive visualization should be included in the main text to support Table 1 and 3. We will add visualization comparisons for LAM and FLAM accordingly. For figure 7, it is more of an ablation study on the aggregator type. We will add quantitative results to strengthen the argument.
>
> > The paper mentions that “human users” were asked to specify latent actions, but it does not clarify which latent actions (e.g., action indices or semantic descriptions) were selected for the visualization in Figure 4.
>
> Yes, we will include the chosen latent actions (action indices in the codebook) in the revision.
>
> > The paper repeatedly claims (Lines 113, 233, 291) that the learned object-centric representations group entities with similar dynamics rather than visual appearance, yet no qualitative evidence is provided beyond overall quantitative metrics. I strongly suggest including visual showcases illustrating this property.
>
> We thank the reviewer for suggesting those experiments to better verify the superiority of our representations. We will add experiments in multigrid to show that our representation can group two entities with different colors but move together into the same factor, while OC+LAM does not.
>
> > Although not an experimental issue, the discussion on slot initialization in Section 5.5 is insightful and should be moved into the method section, which contributes more directly to understanding the model design.
>
> We thank the reviewer for the great suggestion. We put those discussions in the appendix due to page limits. We will revise accordingly.
>
> > Overall, I strongly recommend a thorough revision of the paper’s structure and formatting to make the contributions clearer and ensure that all major claims are properly supported.
>
> Thanks, we will ensure all suggestions above are incorporated. Meanwhile, we notice that most suggestions are easily fixable. We would appreciate it if the reviewer could consider raising the score after we update the paper.
>
> > The title emphasizes “scaling latent action world models,” yet the experiments do not demonstrate any scaling trend with model size or capacity (see also Question 3). To avoid misleading implications, the paper should either include such experiments or adopt a more conservative title reflecting its actual scope.
>
> Thank you for the suggestion. We will revise the paper title, clarifying the scaling refer to the scaling to multiple entities / agent.
>
> > A straightforward baseline could be to factorize latent actions into multiple groups, analogous to how DreamerV2 decomposes latent representations. How would FLAM compare to such a baseline?
>
> To clarify, our LAM implementation has already factorized the latent into multiple action codebooks (e.g., instead of using a single codebook with 16 codes, use 2 codebooks with 4 codes). This modification gives the LAM baseline advantages compared to the original Genie implementation. However, in multigrid and procgen environments, FLAM still outperforms LAM despite that LAM uses a superior implementation.
>
> > The number of slots is known to be a critical hyperparameter in slot attention. Have the authors conducted sensitivity analyses on the number of slots or other key hyperparameters?
>
> We agree that the number of slots K is a critical hyperparemeter. We claim that when we choose K, we set it to a relatively large number so that it serves as an **upper bound** on the number of moving factors, not the number of real entities. As long as the number of factors K > the number of entities N, the world model is relatively accurate. To support our claim, we will do more detailed ablation experiments on this hyperparemeter and report in the revision.

---

### Official Review · Reviewer_M4Wu · 2025-10-31

**Soundness:** 3
**Presentation:** 2
**Contribution:** 2
**Rating:** 2
**Confidence:** 4

**Summary:**

The paper is focused on the problem of video generation with implicitly discovered actions. The work proposed to improve the state of the art -Latent Action Models - by introducing FLMA that has explicit factors and allows the model to model multiple objects with their actions independently, rather then modelling the entire scene. This give FLAM additional representational power and allows for better video controllability.

**Strengths:**

* The idea of introducing this additional degree of freedom in modelling, and allow the model con condition different scene parts on different actions is intuitive and potentially effective
* The idea of using slot attention to implicitly learn object-action slots is neat
* I like that the authors get away with not having to explicitly couple slots at timestamp t and t+1, and still get acceptable results

**Weaknesses:**

* Performance: FLAM achieves inferior results to LAM in PSNR. However, given that the capacity of the representational power of the FLAM is higher, I would expect the opposite. Do the authors know why this could be happening?
* Generalizability: Given that the visual tokenizer and FLMA itself have to be re-trained on every dataset sugges that the latent action space discovered is limited to just the actions in the dataset. This seems to go against the latest fashion where the goal is to train a single model that works everywhere (eg Genie)
* Visual Quality: For the visual tokenizer, why not use the pre-trained VQ-VAE used in diffusion models (for example SDXL [a])? It is clear from Figures 5, 6 that the tokenizer does not work well because the reconstruction show clear checkerboard and blurriness artifacts. Also, to quantify the reconstruction error, it would be helpful to see some metrics (like PSNR) on top of the visualizations.
* While the authors demonstrate the modelling ability of the model, it would be beneficial to use it for downstream applications, like planning. Such experiments are not present in the paper.

[a] Podell et al., SDXL: Improving Latent Diffusion Models for High-Resolution Image Synthesis

**Questions:**

* I am not quite sure what the authors mean by saying that the training objective (5) is qualitatively different from that used in SlotAttention. It is exactly equivalent to the SaVI [a] objective, where the goal is to reconstruct the future frame from the latents. Perhaps this should be explained.

* The authors claim that the slots learn to bind to actions rather than objects. However, is this a desired property? Say, we have a car and a pedestrian turning right, and they may be captured into a single slot, yet, this may lead to “cross-talk” between features of those distinct objects (resulting in undesired artifacts), just because they bind to the same action. It feels like the slots should still be separated by “objectless” and only then bind to actions independently.

* I do not understand how the research question (1) is relevant to the main contribution of this paper - factorization in LAM. It seems to me that it is in fact orthogonal to the paper itself, and this result has been demonstrated before in the video prediction literature. This question would much rather be suitable as model ablation.

[a] Elsayed et al., SAVi++: Towards end-to-end object-centric learning from real-world videos

---

> ### Author Response · Authors · 2025-11-16
> **Response to reviewer's comment**
>
> > Generalizability: Given that the visual tokenizer and FLMA itself have to be re-trained on every dataset suggest that the latent action space discovered is limited to just the actions in the dataset. This seems to go against the latest fashion where the goal is to train a single model that works everywhere (eg Genie)
>
> Prior LAM work (LAPO) trains models per dataset, so our setup follows the standard experimental protocol rather than reflecting a limitation of FLAM. Training a universal latent-action world model requires massive, heterogeneous video corpora and industrial-scale compute (as in Genie); this direction is orthogonal to our contribution and beyond our compute resources (4xA100 GPUs). Conceptually, FLAM’s factorized latent action space is more scalable than monolithic LAMs and can be trained on mixed datasets in future work.
>
> > Visual Quality: For the visual tokenizer, why not use the pre-trained VQ-VAE used in diffusion models (for example SDXL [a])? It is clear from Figures 5, 6 that the tokenizer does not work well because the reconstruction show clear checkerboard and blurriness artifacts. Also, to quantify the reconstruction error, it would be helpful to see some metrics (like PSNR) on top of the visualizations.
>
> * We experimented with several pre-trained VQ-VAE (cosmos, 1d-tokenizer, and Selftok) and unfortunately none of them generalizes to our datasets. So we decided to train our own tokenizer.
> * We thank the reviewer for the great suggestions, we add reconstruction PSNR, which can also serve as an upper bound of prediction performance.
>
> > While the authors demonstrate the modelling ability of the model, it would be beneficial to use it for downstream applications, like planning. Such experiments are not present in the paper.
>
> We trained policy using the pseudo action from the world model. The results were not in the main text due to space limitation. We will add them in the revision to better demonstrate the modeling ability of FLAM.
>
> > I am not quite sure what the authors mean by saying that the training objective (5) is qualitatively different from that used in SlotAttention. It is exactly equivalent to the SaVI [a] objective, where the goal is to reconstruct the future frame from the latents. Perhaps this should be explained.
>
> We thank the reviewer for pointing out this typo. Our intention was not to claim that prediction-based objectives themselves are new, indeed, prior object-centric works such as SaVI also use future-frame prediction rather than reconstruction. We will revise the text to make the following comparison clearer.
>
> What we intended to highlight is the difference between our method against the two-stage approaches (OC+LAM) that first pretrain object-centric representations (OC) with reconstruction/prediction and then train a latent action model (LAM) on top of them. In contrast, FLAM’s end-to-end factorized dynamics learning, where slot extraction, latent action inference, and forward dynamics are jointly optimized under the same prediction objective. In FLAM, factorization is driven by **dynamics independence rather than visual similarity**.
>
> > The authors claim that the slots learn to bind to actions rather than objects. However, is this a desired property? Say, we have a car and a pedestrian turning right, and they may be captured into a single slot, yet, this may lead to “cross-talk” between features of those distinct objects (resulting in undesired artifacts), just because they bind to the same action. It feels like the slots should still be separated by “objectless” and only then bind to actions independently.
>
> * Regarding “cross-walk”: If two entities were captured into one slot and result in inaccurate prediction, it would directly increase the future-frame prediction loss, so the optimization naturally discourages such artifacts. **This is analogous to patch-based video models**, where a single patch often contains multiple objects, yet the prediction loss still leads to clean, artifact-free future frames.
> * Regarding “slot binding”: There is no universally optimal factorization. In many domains, grouping entities by shared dynamics is actually beneficial. For instance, in driving videos, a bus and its passengers, or a car and its carried cargo, form a single rigid dynamical unit. In manipulation tasks, a hand holding an object moves as a single controllable unit. In such cases, separating them into distinct “object-only” slots is unnecessary and can increase learning complexity.
>
> > I do not understand how the research question (1) is relevant to the main contribution of this paper - factorization in LAM. It seems to me that it is in fact orthogonal to the paper itself, and this result has been demonstrated before in the video prediction literature. This question would much rather be suitable as model ablation.
>
> we agree that question 1 is  more of an ablation and will revise the paper accordingly.

---

> > ### Comment · Reviewer_M4Wu · 2025-11-19
> > **Final decision**
> >
> > I thank the authors for taking the time to respond to the questions. Given the amount of modifications needed to the paper, and the outstanding concerns of other reviewers, I keep my score.
> >
> > With some more work and time put in this paper, this will make for a strong submission to the next conference.

---

### Official Review · Reviewer_XeCk · 2025-11-01

**Soundness:** 2
**Presentation:** 2
**Contribution:** 1
**Rating:** 2
**Confidence:** 4

**Summary:**

This paper proposes a factorized way to learn a latent action world model. The paper leverages slot attention to extract features from pixels, and jointly trains the inverse dynamic model and forward dynamic model for world modeling. Experiments are conducted on game datasets and several real-world datasets.

**Strengths:**

- This paper proposes a factorized way for better learning of the latent action world model. It is well-motivated.
- The method is evaluated across diverse datasets, validating the efficacy of the proposed method.

**Weaknesses:**

- The proposed method is not novel, and does not seem to be sound to address the mentioned problem. In the introduction, this paper claims that a single latent action is challenging for tasks that have multiple entities. However, the proposed method proposes to learn multiple actions for each slot $k, k\sim [0,K-1]$, which is not that scalable and clever:
   - 1) The hyperparameter $K$ should be defined and fixed in advance. It can not handle tasks that include diverse entities. Thus, it can not be scaled like Genie to learn from diverse datasets.
   - 2) The slot attention used in this paper can not guarantee that the learned slot is highly related to a specific **actionable** entity.
   - 3) Overall, this paper just integrates previous slot attention to the context of learning a latent action world model.
- Empirical results do not convince me. This paper does not compare with any baselines, except for the variants of the proposed framework. Related work such as [1] and [2] should be discussed and compared with.
- Only PSNR-based metrics are used for evaluation. It is not sufficient to test the video prediction performance. Metrics like FVD and SSIM should be added.
- For Table 2, the advantage of FLAM does not become more obvious as the number of entities increases.
- No qualitative demos are provided.

[1] Adaworld: Learning adaptable world models with latent actions, ICML 2025
[2] Prelar: World model pre-training with learnable action representation, ECCV 2024

**Questions:**

Please address my concerns as detailed in the weaknesses.

---

> ### Author Response · Authors · 2025-11-16
> **Response to reviewer's comment**
>
> We thank the reviewer for the effort engaged in the review phase. The clarifications are as below.
>
> > The hyperparameter K should be defined and fixed in advance. It can not handle tasks that include diverse entities. Thus, it can not be scaled like Genie to learn from diverse datasets.
>
> 1. K is an **upper bound** on the number of moving factors, not the number of real entities. We can set a relatively large K and learn FLAM from diverse datasets.
>
> * Genie also fixes architectural hyperparameters such as spatial token count and latent resolution, which constrain its ability to scale to different scene complexities. K serves an analogous role to these structural decisions; it is not a limiting factor unique to FLAM.
> * To support our claim, we are running experiments with a large K for all datasets and will update results when ready.
> Moreover, Genie does not address multi-agent or multi-entity domains and struggles to model independent, interacting entities, as stated in Genie’s blog.
>
> 2. Moreover, Genie does not address multi-agent or multi-entity domains and struggles to model independent, interacting entities, as stated in Genie’s [blog](https://deepmind.google/blog/genie-3-a-new-frontier-for-world-models/#limitations).
>
> > The slot attention used in this paper can not guarantee that the learned slot is highly related to a specific **actionable** entity.
>
> We note that, in general, no unsupervised representation learning method (even object-centric works such as MONet or Slot Attention, etc.) can theoretically guarantee slot–entity alignment. What matters is the inductive bias introduced by the learning objective.
> In FLAM, the inverse/forward dynamics prediction loss explicitly encourages each slot to capture a region with independent dynamics, which empirically yields consistent alignment with actionable entities, as shown in Appendix F Fig. 9.
>
> > Overall, this paper just integrates previous slot attention to the context of learning a latent action world model.
>
> While FLAM uses slot attention as a factorizer, the central contribution is not architectural integration. FLAM introduces a **factorized latent action space**, a **shared per-factor action codebook**, and **factorized IDM/FDM dynamics trained with prediction rather than reconstruction**. These elements jointly enable linear scaling with entity count, which monolithic LAMs (including Genie) cannot achieve.
>
> > Empirical results do not convince me. This paper does not compare with any baselines, except for the variants of the proposed framework. Related work such as [1] and [2] should be discussed and compared with.
>
> **LAM is a revised baseline, not a variant of our proposed framework**. It is actually LAPO [1] which we believe Prelar[2] is highly similar to,, but we revised the extraction part of LAPO to align with FLAM for fair comparison and named it LAM. We will revise the name to better highlight the baseline. For Adaworld[1], we agree it is a SOTA world model. However, we think **it is not a fair comparison considering Adaworld is based on diffusion architecture**, we would not be able to if the performance difference origins from factorization design or architecture difference.
>
> References:
> [3] Schmidt, Dominik & Jiang, Minqi. Learning to Act without Actions. In Proceedings of the International Conference on Learning Representations (ICLR 2024).
>
> > No qualitative demos are provided.
> Please refer to appendix F for qualitative demos.

---

### Official Review · Reviewer_wUV1 · 2025-11-01

**Soundness:** 2
**Presentation:** 3
**Contribution:** 3
**Rating:** 4
**Confidence:** 4

**Summary:**

This paper proposes an approach that factorises the latent action space with a shared codebook to enable better modelling of videos containing multiple entities. The approach involves using slot attention to extract these factorised latent actions and decompose the action space in this manner to achieve linear instead of combinatorial actions. This enables generating diverse videos from the same initial frame.

**Strengths:**

- The paper is well-written and proposes a neat trick to tackle video generation in cases where there are multiple entities in the video through latent action models. This is an important problem to work on and factorisation of the action space in this manner provides an interpretable solution.
- The promise of the method in generating diverse videos is the biggest strength of the paper.
- The approach is described well and grounded in prior literature.

**Weaknesses:**

- The lack of a limitations section is a key weakness in the paper: increasing diversity comes at the expense of inconsistency in the video (example in the last point in this section). This is not something that this method addresses and is a limitation, but it would be helpful to the community to have a discussion on this subject so that future work can combine the benefits of factorisation with other work on consistency and fidelity in video generation.
- One critical section that does not appear in the main text is the analysis of the results comparing this approach with the baselines in terms of the benchmark and selection of hyperparameters. While lines 405-408 mention the fact that the number of factors need to be carefully chosen considering the complexity of the scenes, there is no follow-up analysis indicating how this choice is made, and what it implies in terms of the practicality of the method. If the hyperparameter $N$ has to be selected keeping in mind the number of entities appearing in every single video, that makes the proposed method impractical. In addition to this, there is no discussion or evidence on why the number of latent factors was determined to be the reason for the drop in performance using FLAM compared to LAM.
- In Tables 1 and 3, the results for OC+LAM seem to be quite unexpected, given that one key difference between OC+LAM and FLAM is the object-wise reconstruction loss. The additional reconstruction loss should ideally help with PSNR, if not $\Delta$PSNR, but we see consistently worse metrics for OC+LAM compared to both LAM and FLAM. This inconsistency seems quite unsatisfactory and further discussion is warranted, especially since this is an important baseline and closer to FLAM conceptually than LAM.
- In figure 4, it appears that of the two generated next states with the same initial frame, both next frames have certain impossible changes (the white car disappearing, an extra lane being added, etc.). I understand that diverse next frame generation would be much easier with a factored latent space but it is also equally important to discuss any guardrails that this approach has against unrealistic video generation, and if not, the limitations of this approach in that regard.

**Questions:**

- While extracting factored latents, is this number $N$ fixed? If not, is there an ablation over the number of slots and the impact this has on performance and diversity of generations? Note that this is not the same question as is being addressed in Table 2, if I understand correctly. Table 2 measures the impact of having multiple entities in the video on reconstruction metrics. My question is about the hyperparameter, not the environment parameters: in a video where the number of entities is unknown, how would you set the value $N$? Is there an ablation on this in the following scenarios: 1) $N$ < number of entities in the video, in which case the model might extract different factors in each frame, and 2) $N$ > number of entities in the video, in which case some of the factors might be noisy and potentially adversely impact performance.
- Just as table 2 contains PSNR results as the number of entities in the videos is scaled up, is there a similar table for $\Delta$PSNR?
- If we were to think intuitively, maximising diversity through factorisation would imply that reconstruction metrics might be lower. However, the results in table 1 show that the PSNR is higher than LAM and OC-LAM, which seems counter-intuitive. Could the authors give an explanation for why this is the case?
- A key fact the paper mentions is that this approach is different from an object-centric representation-based approach in that groups with similar dynamics can be clubbed because there is no reliance on superficial visual features. However, in the environments studied, there is no such environment that showcases this key difference. Are there results on such environments in which entities with different visual properties still behave the same way and are thus advantaged from the factored approach versus the object-centric approach?

---

> ### Author Response · Authors · 2025-11-16
> **Response to reviewer's comment (1/2)**
>
> We genuinely thank the reviewer for their constructive feedback. Regarding the concerns and questions, we provide the the response as below and will incorporate modification accordingly in the revision.
>
> > The lack of a limitations section is a key weakness in the paper: increasing diversity comes at the expense of inconsistency in the video (example in the last point in this section). This is not something that this method addresses and is a limitation, but it would be helpful to the community to have a discussion on this subject ...
>
> We thank the reviewer for bringing up this limitation. We agree inconsistency is an unaddressed point, and will add a limitation section to discuss this more.
>
> > One critical section that does not appear in the main text is the analysis of the results comparing this approach with the baselines in terms of the benchmark and selection of hyperparameters. While lines 405-408 mention the fact that the number of factors need to be carefully chosen considering the complexity of the scenes, there is no follow-up analysis indicating how this choice is made, and what it implies in terms of the practicality of the method. If the hyperparameter has to be selected keeping in mind the number of entities appearing in every single video, that makes the proposed method impractical. In addition to this, there is no discussion or evidence on why the number of latent factors was determined to be the reason for the drop in performance using FLAM compared to LAM.
>
> Hyperparameter K is not avoidable in work that involves factorization or composition, e.g. SlotAttention. K is an **upper bound** on the number of factors, not the number of real entities. We can **set a relatively large K**, as long as the number of factors K > the number of entities N, the world model is relatively accurate. When we write lines 405-408, we are trying to emphasize that the choice of K should be relatively large. The phrasing “carefully chosen” is misleading, we will rephrase it in the revision.
>
> We point to the increased number of factors as the reason for the drop in controllability performance using FLAM compared to LAM is because the $\Delta$ PSNR rely on a term where latent action is randomly sampled from a categorical distribution. When K is too large, the sampled latent action could have higher probability manipulating a noisy factor that is not related to an entity and damage the controllability.
>
> > In Tables 1 and 3, the results for OC+LAM seem to be quite unexpected, given that one key difference between OC+LAM and FLAM is the object-wise reconstruction loss. The additional reconstruction loss should ideally help with PSNR, if not $\Delta$ PSNR, but we see consistently worse metrics for OC+LAM compared to both LAM and FLAM. This inconsistency seems quite unsatisfactory and further discussion is warranted, especially since this is an important baseline and closer to FLAM conceptually than LAM.
>
> The consistently worse performance of OC+LAM is actually expected and validates our design choice of not using object-centric representation. We clarify that, the reconstruction loss that OC+LAM uses to learn object-centric representations is calculated on the current frame, while the PSNR metrics reported are on **predicted future frames (where forward dynamics is used)**. The experiments show that, as stated in Lines 412-413, the reconstruction loss used in OC+LAM leads to slot representations **beneficial for visual reconstruction rather than dynamics learning** . FLAM without this loss instead extracts slot representations that better capture dynamics, therefore has higher **PSNR on predicted future frames**.
>
> > In figure 4, it appears that of the two generated next states with the same initial frame, both next frames have certain impossible changes (the white car disappearing, an extra lane being added, etc.). ... it is also equally important to discuss any guardrails that this approach has against unrealistic video generation, and if not, the limitations of this approach in that regard.
>
> All video generation models could not avoid unrealistic generation to some extent. The problem can be solved through training on a larger scale of data, injecting more prior knowledge etc. However, this is not the focus of our paper. We acknowledge that better guardrails are necessary, and would add discussion about this limitation to our revision.
>
> When mentioning diverse video generation, we are claiming we are more diverse than non-factored models in that ** our model can manipulate each agent separately** in video generation, while non-factored models would only be able to manipulate the entire scene together. While this effect might be less visually apparent in the autonomous driving case we show, we show that in the multigrid environment, we can generate frames that only one entity is behavioring differently when all other entities remain unchanged.

---

> ### Author Response · Authors · 2025-11-16
> **Response to reviewer's comment (2/2)**
>
> > While extracting factored latents, is this number fixed? If not, is there an ablation over the number of slots and the impact this has on performance and diversity of generations?... in a video where the number of entities is unknown, how would you set the value ? Is there an ablation on this in the following scenarios: 1)  < number of entities in the video, in which case the model might extract different factors in each frame, and 2)  > number of entities in the video, in which case some of the factors might be noisy and potentially adversely impact performance.
>
> Yes, the number of factors K is fixed for each environment, but would be different between different environments. For simulation environments like multigrid and procgen where the number of entities are known, we set K to be the same as the number of entities. For videos where the number of entities are unknown, we set a relatively large K (K>number of entities N) to ensure the capacity of our model. In this way, some factors indeed would be noisy and not related to any entity, but they would have relatively slight impact to the performance given our aggregator design. We will add ablation experiments to showcase the circumstances when K\<N and K\>N.
>
> > Just as table 2 contains PSNR results as the number of entities in the videos is scaled up, is there a similar table for delta PSNR?
>
> We will add the delta PSNR for table 2 in the revision.
>
> > If we were to think intuitively, maximizing diversity through factorization would imply that reconstruction metrics might be lower. However, the results in table 1 show that the PSNR is higher than LAM and OC-LAM, which seems counter-intuitive. Could the authors give an explanation for why this is the case?
>
> We believe the ambiguity comes from how evaluation setups differ between model prediction and video generation rollout. In Table 1, ** PSNR are reported for prediction (using inferred latent actions)**, not rollouts for video generation (using randomly sampled latent actions).
>
> We would appreciate it if the reviewer can clarify a bit more about what “diversity” is referring to here. We mentioned diversity in controllable video generation, one application area where FLAM can be used, as an outcome of factorization. However, in the learning stage, maximizing diversity is never used as a learning signal for the world model. Factorization is used for learning a more structured world model with the goal of **capturing the dynamics more accurately**, through **disentangling the dynamics of multiple entities**. Therefore higher PSNR would align with the intuition.
>
> > A key fact the paper mentions is that this approach is different from an object-centric representation-based approach in that groups with similar dynamics can be clubbed because there is no reliance on superficial visual features. However, in the environments studied, there is no such environment that showcases this key difference. Are there results on such environments in which entities with different visual properties still behave the same way and are thus advantaged from the factored approach versus the object-centric approach?
>
> We thank the reviewer for suggesting those experiments to better verify the superiority of our representations. We will add experiments in multigrid to show that our representation can group two entities with different colors but move together into the same factor, while OC+LAM does not.

---

### Meta-Review · Area_Chair_omoj · 2026-01-04

**Summary:**

The paper proposes FLAM, a framework for learning latent action world models using factorized states to handle multi-entity scenarios. While the reviewers recognized the novelty of applying object-centric concepts to latent action models and appreciated the evaluation across several datasets, the consensus is that the paper is not ready for publication due to several critical concerns in this round, including a lack of evidence for the core mechanism and structural/presentation/positioning problems. Given the unsubstantiated core claims, it is recommended for rejection in its current form.

**Reviewer Concerns:**

Outstanding concerns: Most of the concerns raised by reviewers are not properly addressed during the rebuttal stage, including the following major points below.
1. The paper claims FLAM groups entities based on dynamics rather than visual appearance, but the reviewer noted a lack of qualitative evidence (visualizations) to support this. The rebuttal only promised future experiments rather than providing them.
2. Critical experimental evidence supporting design choices (specifically Figure 7 regarding Regression vs. Classification) is described in the appendix rather than the main text, which makes the paper not self-contained. Additionally, some methodological details are misplaced in the experiment section.
3. A reviewer pointed out that "Scaling" implies scaling laws (model/data size), whereas the paper focuses on multi-entity complexity. The authors admitted this was misleading but the issue remains in the current manuscript.
4. There is not adequate analysis on how the number of slots affects performance, a critical hyperparameter for the factorization approach.

**Reviewer Scores:**

It is likely that all four reviewers maintain their scores of 4-2-2-2 after the rebuttal.

---

### Decision · Program_Chairs · 2026-01-26

Reject